# Marginalized Average Attentional Network for Weakly-Supervised Learning

**Yuan Yuan[12], Yueming Lyu[3], Xi Shen[4], Ivor W. Tsang[3] & Dit-Yan Yeung[1]**
[1]Hong Kong University of Science and Technology, [2]Alibaba Group
[3]University of Technology Sydney, [4]Ecole des Ponts ParisTech

## Abstract

In weakly-supervised temporal action localization, previous works have failed to locate dense and integral regions for each entire action due to the overestimation of the most salient regions. To alleviate this issue, we propose a marginalized average attentional network (MAAN) to suppress the dominant response of the most salient regions in a principled manner. The MAAN employs a novel marginalized average aggregation (MAA) module and learns a set of latent discriminative probabilities in an end-to-end fashion. MAA samples multiple subsets from the video snippet features according to a set of latent discriminative probabilities and takes the expectation over all the averaged subset features. Theoretically, we prove that the MAA module with learned latent discriminative probabilities successfully reduces the difference in responses between the most salient regions and the others. Therefore, MAAN is able to generate better class activation sequences and identify dense and integral action regions in the videos. Moreover, we propose a fast algorithm to reduce the complexity of constructing MAA from $O(2^T)$ to $O(T^2)$. Extensive experiments on two large-scale video datasets show that our MAAN achieves a superior performance on weakly-supervised temporal action localization.

## 1 Introduction

Weakly-supervised temporal action localization has been of interest to the community recently. The setting is to train a model with solely video-level class labels, and to predict both the class and the temporal boundary of each action instance at the test time. The major challenge in the weakly-supervised localization problem is to find the right way to express and infer the underlying location information with only the video-level class labels. Traditionally, this is achieved by explicitly sampling several possible instances with different locations and durations (Bilen & Vedaldi, 2016; Kantorov et al., 2016; Zhang et al., 2017). The instance-level classifiers would then be trained through multiple instances learning (Cinbis et al., 2017; Yuan et al., 2017a) or curriculum learning (Bengio et al., 2009). However, the length of actions and videos varies too much such that the number of instance proposals for each video varies a lot and it can also be huge. As a result, traditional methods based on instance proposals become infeasible in many cases.

Recent research, however, has pivoted to acquire the location information by generating the class activation sequence (CAS) directly (Nguyen et al., 2018), which produces the classification score sequence of being each action for each snippet over time. The CAS along the 1D temporal dimension for a video is inspired by the class activation map (CAM) (Zhou et al., 2016a; 2014; Pinheiro & Collobert, 2015; Oquab et al., 2015) in weakly-supervised object detection. The CAM-based models have shown that despite being trained on image-level labels, convolutional neural networks (CNNs) have the remarkable ability to localize objects. Similar to object detection, the basic idea behind CAS-based methods for action localization in the training is to sample the non-overlapping snippets from a video, then to aggregate the snippet-level features into a video-level feature, and finally to yield a video-level class prediction. During testing, the model generates a CAS for each class that identifies the discriminative action regions, and then applies a threshold on the CAS to localize each action instance in terms of the start time and the end time.

In CAS-based methods, the feature aggregator that aggregates multiple snippet-level features into a video-level feature is the critical building block of weakly-supervised neural networks. A model's

ability to capture the location information of an action is primarily determined by the design of the aggregators. While using the global average pooling over a full image or across the video snippets has shown great promise in identifying the discriminative regions (Zhou et al., 2016a; 2014; Pinheiro & Collobert, 2015; Oquab et al., 2015), treating each pixel or snippet equally loses the opportunity to benefit from several more essential parts. Some recent works (Nguyen et al., 2018; Zhu et al., 2017) have tried to learn attentional weights for different snippets to compute a weighted sum as the aggregated feature. However, they suffer from the weights being easily dominated by only a few most salient snippets.

In general, models trained with only video-level class labels tend to be easily responsive to small and sparse discriminative regions from the snippets of interest. This deviates from the objective of the localization task that is to locate dense and integral regions for each entire action. To mitigate this gap and reduce the effect of the domination by the most salient regions, several heuristic tricks have been proposed to apply to existing models. For example, (Wei et al., 2017; Zhang et al., 2018b) attempt to heuristically erase the most salient regions predicted by the model which are currently being mined, and force the network to attend other salient regions in the remaining regions by forwarding the model several times. However, the heuristic multiple-run model is not end-to-end trainable. It is the ensemble of multiple-run mined regions but not the single model's own ability that learns the entire action regions. "Hide-and-seek"(Singh & Lee, 2017) randomly masks out some regions of the input during training, enforcing the model to localize other salient regions when the most salient regions happen to be masked out. However, all the input regions are masked out with the same probability due to the uniform prior, and it is very likely that most of the time it is the background that is being masked out. A detailed discussion about related works can be found in Appendix D.

To this end, we propose the marginalized average attentional network (MAAN) to alleviate the issue raised by the domination of the most salient region in an end-to-end fashion for weakly-supervised action localization. Specifically, MAAN suppresses the action prediction response of the most salient regions by employing marginalized average aggregation (MAA) and learning the latent discriminative probability in a principled manner. Unlike the previous attentional pooling aggregator which calculates the weighted sum with attention weights, MAA first samples a subset of features according to their latent discriminative probabilities, and then calculates the average of these sampled features. Finally, MAA takes the expectation (marginalization) of the average aggregated subset features over all the possible subsets to achieve the final aggregation. As a result, MAA not only alleviates the domination by the most salient regions, but also maintains the scale of the aggregated feature within a reasonable range. We theoretically prove that, with the MAA, the learned latent discriminative probability indeed reduces the difference of response between the most salient regions and the others. Therefore, MAAN can identify more dense and integral regions for each action. Moreover, since enumerating all the possible subsets is exponentially expensive, we further propose a fast iterative algorithm to reduce the complexity of the expectation calculation procedure and provide a theoretical analysis. Furthermore, MAAN is easy to train in an end-to-end fashion since all the components of the network are differentiable. Extensive experiments on two large-scale video datasets show that MAAN consistently outperforms the baseline models and achieves superior performance on weakly-supervised temporal action localization.

In summary, our main contributions include: (1) a novel end-to-end trainable marginalized average attentional network (MAAN) with a marginalized average aggregation (MAA) module in the weakly-supervised setting; (2) theoretical analysis of the properties of MAA and an explanation of the reasons MAAN alleviates the issue raised by the domination of the most salient regions; (3) a fast iterative algorithm that can effectively reduce the computational complexity of MAA; and (4) a superior performance on two benchmark video datasets, THUMOS14 and ActivityNet1.3, on the weakly-supervised temporal action localization.

## 2 MARGINALIZED AVERAGE ATTENTIONAL NETWORK

In this section, we describe our proposed MAAN for weakly-supervised temporal action localization. We first derive the formulation of the feature aggregation module in MAAN as a MAA procedure in Sec. 2.1. Then, we study the properties of MAA in Sec. 2.2, and present our fast iterative computation algorithm for MAA construction in Sec. 2.3. Finally, we describe our network architecture that

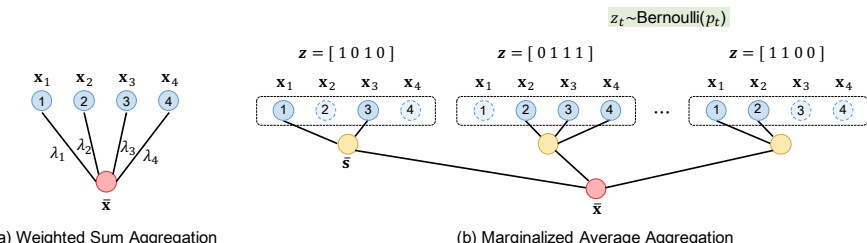

Figure 1: An illustration of the weighted sum aggregation and the marginalized average aggregation.

incorporates MAA, and introduce the corresponding inference process on weakly-supervised temporal action localization in Sec. 2.4.

## 2.1 MARGINALIZED AVERAGE AGGREGATION

Let $\{\mathbf{x}_1, \mathbf{x}_2, \cdots \mathbf{x}_T\}$ denote the set of snippet-level features to be aggregated, where $\mathbf{x}_t \in \mathbb{R}^m$ is the $m$ dimensional feature representation extracted from a video snippet centered at time $t$, and $T$ is the total number of sampled video snippets. The conventional attentional weighted sum pooling aggregates the input snippet-level features into a video-level representation $\overline{\mathbf{x}}$. Denote the set of attentional weights corresponding to the snippet-level features as $\{\lambda_1, \lambda_2, \cdots \lambda_T\}$, where $\lambda_t$ is a scalar attentional weight for $\mathbf{x}_t$. Then the aggregated video-level representation is given by

$$\overline{\mathbf{x}} = \sum_{t=1}^{T} \lambda_t \mathbf{x}_t, \tag{1}$$

as illustrated in Figure 1 (a). Different from the conventional aggregation mechanism, the proposed MAA module aggregates the features by firstly generating a set of binary indicators to determine whether a snippet should be sampled or not. The model then computes the average aggregation of these sampled snippet-level representations. Lastly, the model computes the expectation (marginalization) of the aggregated average feature for all the possible subsets, and obtains the proposed marginalized average aggregated feature. Formally, in the proposed MAA module, we first define a set of probabilities $\{p_1, p_2, \cdots p_T\}$, where each $p_t \in [0, 1]$ is a scalar corresponding to $\mathbf{x_t}$, similar to the notation $\lambda_t$ mentioned previously. We then sample a set of random variables $\{z_1, z_2, \cdots z_T\}$, where $z_t \sim Bernoulli(p_t)$, i.e., $z_t \in \{0, 1\}$ with probability $P(z_t = 1) = p_t$. The sampled set is used to represent the subset selection of snippet-level features, in which $z_t = 1$ indicates $\mathbf{x}_t$ is selected, otherwise not. Therefore, the average aggregation of the sampled subset of snipped-level representations is given by $\overline{\mathbf{s}} = \sum_{i=1}^{T} z_i \mathbf{x}_i / \sum_{i=1}^{T} z_i$ , and our proposed aggregated feature, defined as the expectation of all the possible subset-level average aggregated representations, is given by

$$\overline{\mathbf{x}} = \mathbb{E}[\overline{\mathbf{s}}] = \mathbb{E}\left[\frac{\sum_{i=1}^{T} z_i \mathbf{x}_i}{\sum_{i=1}^{T} z_i}\right], \tag{2}$$

which is illustrated in Figure 1 (b).

## 2.2 PARTIAL ORDER PRESERVATION AND DOMINANT RESPONSE SUPPRESSION

Direct learning and prediction with the attention weights $\lambda$ in Eq. (1) in weakly-supervised action localization leads to an over-response in the most salient regions. The MAA in Eq. (2) has two properties that alleviate the domination effect of the most salient regions. First, the partial order preservation property, i.e., the latent discriminative probabilities preserve the partial order with respect to their attention weights. Second, the dominant response suppression property, i.e., the differences in the latent discriminative probabilities between the most salient items and others are smaller than the differences between their attention weights. The partial order preservation property guarantees that it does not mix up the action and non-action snippets by assigning a high latent discriminative probability to a snippet with low response. The dominant response suppression property reduces

the dominant effect of the most salient regions and encourages the identification of dense and more integral action regions. Formally, we present the two properties in Proposition 1 and Proposition 2, respectively. Detailed proofs can be found in Appendix A and Appendix B respectively.

**Proposition 1.** *Let $z_i \sim Bernoulli(p_i)$ for $i \in \{1, ..., T\}$. Then for $T \geq 2$, Eq. (3) holds true, and $p_i \geq p_j \Leftrightarrow c_i \geq c_j \Leftrightarrow \lambda_i \geq \lambda_j$.*

$$\mathbb{E}\left[\frac{\sum_{i=1}^{T} z_i \mathbf{x}_i}{\sum_{i=1}^{T} z_i}\right] = \sum_{i=1}^{T} c_i p_i \mathbf{x}_i = \sum_{i=1}^{T} \lambda_i \mathbf{x}_i, \tag{3}$$

*where $c_i = \mathbb{E}\left[1/(1 + \sum_{k=1, k \neq i}^{T} z_k)\right]$ and $\lambda_i = c_i p_i$ for $i \in \{1, ..., T\}$.*

Proposition 1 shows that the latent discriminative probabilities $\{p_i\}$ preserve the partial order of the attention weights $\{\lambda_i\}$. This means that a large attention weight corresponds to a large discriminative probability, which guarantees that the latent discriminative probabilities preserve the ranking of the action prediction response. Eq. (3) can be seen as a factorization of the attention weight $\lambda_i$ into the multiplication of two components, $p_i$ and $c_i$, for $i \in \{1, ..., T\}$. $p_i$ is the latent discriminative probability related to the feature of snippet $i$ itself. The factor $c_i$ captures the contextual information of snippet $i$ from the other snippets. This factorization can be considered to be introducing structural information into the aggregation. Factor $c_i$ can be considered as performing a structural regularization for learning the latent discriminative probabilities $p_i$ for $i \in \{1, ..., T\}$, as well as for learning a more informative aggregation.

**Proposition 2.** *Let $z_i \sim Bernoulli(p_i)$ for $i \in \{1, ..., T\}$. Denote $c_i = \mathbb{E}\left[1/(1 + \sum_{k=1, k \neq i}^{T} z_k)\right]$ and $\lambda_i = c_i p_i$ for $i \in \{1, ..., T\}$. Denote $\mathcal{I} = \left\{i \,\middle|\, c_i \geq 1/(\sum_{t=1}^{T} p_t)\right\}$ as an index set. Then $\mathcal{I} \neq \emptyset$ and for $\forall i \in \mathcal{I}$, $\forall j \in \{1, ..., T\}$ inequality (4) holds true.*

$$\left|\frac{p_i}{\sum_{t=1}^{T} p_t} - \frac{p_j}{\sum_{t=1}^{T} p_t}\right| \leq \left|\frac{\lambda_i}{\sum_{t=1}^{T} \lambda_t} - \frac{\lambda_j}{\sum_{t=1}^{T} \lambda_t}\right| \tag{4}$$

The index set $\mathcal{I}$ can be viewed as the most salient features set. Proposition 2 shows that the difference between the normalized latent discriminative probabilities of the most salient regions and others is smaller than the difference between their attention weights. It means that the prediction for each snippet using the latent discriminative probability can reduce the gap between the most salient featuress and the others compared to conventional methods that are based on attention weights. Thus, MAAN suppresses the dominant responses of the most salient featuress and encourages it to identify dense and more integral action regions.

Directly learning the attention weights $\lambda$ leans to an over response to the most salient region in weakly-supervised temporal localization. Namely, the attention weights for only a few snippets are too large and dominate the others, while attention weights for most of the other snippets that also belong to the true action are underestimated. Proposition 2 shows that latent discriminative probabilities are able to reduce the gap between the most salient features and the others compared to the attention weights. Thus, by employing the latent discriminative probabilities for prediction instead of the attention weights, our method can alleviate the dominant effect of the most salient region in weakly-supervised temporal localization.

## 2.3 RECURRENT FAST COMPUTATION

Given a video containing $T$ snippet-level representations, there are $2^T$ possible configurations for the subset selection. Directly summing up all the $2^T$ configurations to calculate $\bar{\mathbf{x}}$ has a complexity of $O(2^T)$. In order to reduce the exponential complexity, we propose an iterative method to calculate $\bar{\mathbf{x}}$ with $O(T^2)$ complexity. Let us denote the aggregated feature of $\{\mathbf{x}_1, \mathbf{x}_2, \cdots \mathbf{x}_t\}$ with length $t$ as $\mathbf{h}_t$, and denote $\mathbf{Y_t} = \sum_{i=1}^{t} z_i \mathbf{x}_i$ and $Z_t = \sum_{i=1}^{t} z_i$ for simplicity, then we have a set of

$$\mathbf{h}_t = \mathbb{E}\left[\frac{\sum_{i=1}^{t} z_i \mathbf{x}_i}{\sum_{i=1}^{t} z_i}\right] = \mathbb{E}\left[\frac{\mathbf{Y_t}}{Z_t}\right], t \in \{1, 2, \cdots, T\}, \tag{5}$$

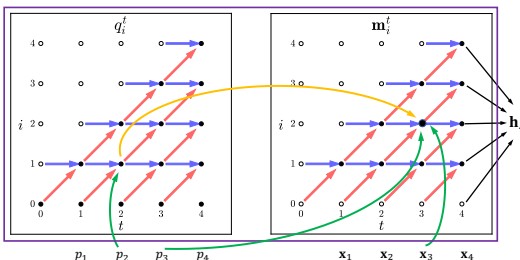

Figure 2: The purple box demonstrates the marginalized average aggregation module, where the inputs are $\{p_i\}_{i=1}^{4}$ and $\{\mathbf{x_i}\}_{i=1}^{4}$ and the output is $\mathbf{h_4}$. The two black boxes demonstrate the computation graphs of $q_i^t$ and $\mathbf{m_i^t}$, respectively. The black hollow point indicates its value is 0, while the value of the black solid point is non-zero. $q_0^0$ is initialized as 1.

and the aggregated feature of $\{\mathbf{x}_1, \mathbf{x}_2, \cdots \mathbf{x}_T\}$ can be obtained as $\overline{\mathbf{x}} = \mathbf{h}_T$. In Eq. (5), $Z_t$ is the summation of all the $z_i$, which indicates the number of elements selected in the subset. Although there are $2^t$ distinct configurations for $\{z_1, z_2, \cdots z_t\}$, it has only $t + 1$ distinct values for $Z_t$, i.e. $0, 1, \cdots, t$. Therefore, we can divide all the $2^t$ distinct configurations into $t + 1$ groups, where the configurations sharing with the same $Z_t$ fall into the same group. Then the expectation $\mathbf{h}_t$ can be calculated as the summation of the $t + 1$ parts. That is, $\mathbf{h}_t = \mathbb{E}\left[\mathbb{E}\left[\frac{\mathbf{Y_t}}{Z_t} \,\middle|\, Z_t = i\right]\right] = \sum_{i=0}^{t} \mathbf{m}_i^t$, where the $\mathbf{m}_i^t$, indicating the $i^{th}$ part of $\mathbf{h}_t$ for group $Z_t = i$, is shown in Eq. (6).

$$\mathbf{m}_i^t = P\left(Z_t = i\right) \mathbb{E}\left[\frac{\mathbf{Y_t}}{Z_t} \,\middle|\, Z_t = i\right]. \tag{6}$$

In order to calculate $\mathbf{h}_{t+1} = \sum_{i=0}^{t+1} \mathbf{m}_i^{t+1}$, given $\mathbf{m}_i^t$, $i \in \{0, \cdots, t\}$, we can calculate $\mathbf{m}_i^{t+1}$, $i \in \{0, 1, \cdots, t + 1\}$ recurrently. The key idea here is that $\mathbf{m}_i^{t+1}$ comes from two cases: if $z_{t+1} = 0$, then $\mathbf{m}_i^{t+1}$ is the same as $\mathbf{m}_i^t$; if $z_{t+1} = 1$, then $\mathbf{m}_i^{t+1}$ is the weighted average of $\mathbf{m}_{i-1}^t$ and $\mathbf{x}_{t+1}$. The latter case is also related to the probability $P\left(Z_t = i - 1\right)$. By denoting $q_{i-1}^t = P\left(Z_t = i - 1\right)$ for simplicity, we can obtain $\mathbf{m}_i^{t+1}$ as a function of several elements:

$$\mathbf{m}_i^{t+1} = f(\mathbf{m}_{i-1}^t, \mathbf{m}_i^t, \mathbf{x}_{t+1}, p_{t+1}, q_{i-1}^t). \tag{7}$$

Similarly, the computation of $q_i^{t+1} = P\left(Z_{t+1} = i\right)$ comes from two cases: the probability of selecting $i - 1$ items from the first $t$ items and selecting the $(t + 1)^{th}$ item, i.e., $q_{i-1}^t p_{t+1}$; and the probability of selecting $i$ items all from the first $t$ items and not selecting the $(t + 1)^{th}$ item, i.e., $q_i^t \left(1 - p_{t+1}\right)$. We derive the function of $\mathbf{m}_i^{t+1}$ and $q_i^{t+1}$ in Proposition 3. Detailed proofs can be found in Appendix C.

**Proposition 3.** *Let $z_t \sim Bernoulli(p_t)$, $Z_t = \sum\limits_{i=1}^{t} z_i$ and $\mathbf{Y_t} = \sum\limits_{i=1}^{t} z_i \mathbf{x}_i$ for $t \in \{1, ..., T\}$. Define $\mathbf{m}_i^t$, $i \in \{0, \cdots, t\}$ as Eq. (6) and $q_i^t = P\left(Z_t = i\right)$, then $\mathbf{m}_i^{t+1}$ $i \in \{0, 1, \cdots, t + 1\}$ can be obtained recurrently by Eq. (8) and Eq. (9).*

$$\mathbf{m}_i^{t+1} = p_{t+1}\left(b_{i-1}\mathbf{m}_{i-1}^t + (1 - b_{i-1})q_{i-1}^t\mathbf{x}_{t+1}\right) + (1 - p_{t+1})\mathbf{m}_i^t, \tag{8}$$

$$q_i^{t+1} = p_{t+1}q_{i-1}^t + (1 - p_{t+1})q_i^t, \tag{9}$$

*where $b_i = \frac{i}{i+1}$, $q_{-1}^t = 0$, $q_{t+1}^t = 0$, $q_0^0 = 1$, $\mathbf{m}_0^t = \mathbf{0}$, and $\mathbf{m}_{t+1}^t = \mathbf{0}$.*

Proposition 3 provides a recurrent formula to calculate $\mathbf{m}_i^t$. With this recurrent formula, we calculate the aggregation $\mathbf{h}_T$ by iteratively calculating $\mathbf{m}_i^t$ from $i = 1$ to $t$ and $t = 1$ to $T$. Therefore, we can obtain the aggregated feature of $\{\mathbf{x}_1, \mathbf{x}_2, \cdots \mathbf{x}_T\}$ as $\overline{\mathbf{x}} = \mathbf{h}_T = \sum_{i=0}^{T} \mathbf{m}_i^T$. The iterative computation procedure is summarized in Algorithm 1 in Appendix E. The time complexity is $O(T^2)$.

With the fast iterative algorithm in Algorithm 1, the MAA becomes practical for end-to-end training. A demonstration of the computation graph for $q_i^{t+1}$ in Eq. (9) and $\mathbf{m}_i^{t+1}$ in Eq. (8) is presented in the left and right-hand sides of Figure 2, respectively. From Figure 2, we can see clearly that, to compute $\mathbf{m}_2^3$ (the big black node on the right), it needs $\mathbf{m}_1^2$, $\mathbf{m}_2^2$, $\mathbf{x_3}$, $p_3$, and $q_1^2$. The MAA can be easily implemented as a subnetwork for end-to-end training and can be used to replace the operation of other feature aggregators.

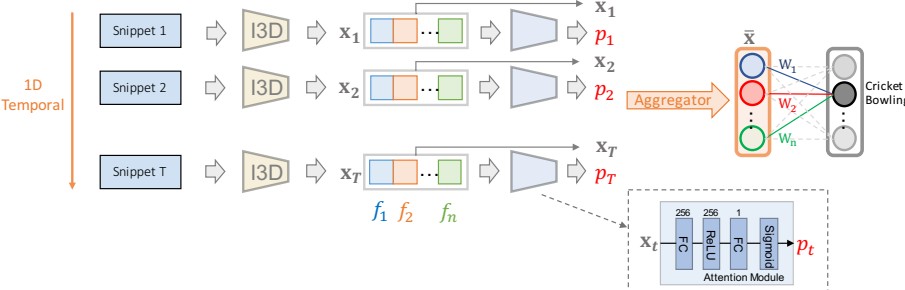

Figure 3: Network architecture for the weakly-supervised action localization.

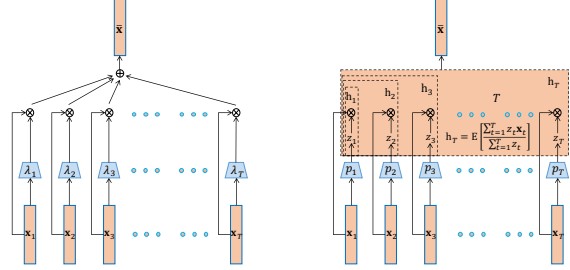

Figure 4: The feature aggregators used in STPN and MAAN.

## 2.4 NETWORK ARCHITECTURE AND TEMPORAL ACTION LOCALIZATION

**Network Architecture**: We now describe the network architecture that employs the MAA module described above for weakly-supervised temporal action localization. We start from a previous state-of-the-art base architecture, the sparse temporal pooling network (STPN) (Nguyen et al., 2018). As shown in Figure 3, it first divides the input video into several non-overlapped snippets and extracts the I3D (Carreira & Zisserman, 2017) feature for each snippet. Each snippet-level feature is then fed to an attention module to generate an attention weight between 0 and 1. STPN then uses a feature aggregator to calculate a weighted sum of the snippet-level features with these class-agnostic attention weights to create a video-level representation, as shown on the left in Figure 4. The video-level representation is then passed through an FC layer followed by a sigmoid layer to obtain class scores. Our MAAN uses the attention module to generate the latent discriminative probability $p_t$ and replaces the feature aggregator from the weighted sum aggregation by the proposed marginalized average aggregation, which is demonstrated on the right in Figure 4.

**Training with video-level class labels**: Formally, the model first performs aggregation of the snippet-level features (i.e. $\mathbf{x}_1, \mathbf{x}_2, \cdots \mathbf{x}_T$ ) to obtain the video-level representation $\bar{\mathbf{x}}$ ( $\bar{\mathbf{x}} = \mathbb{E}[\sum_{i=1}^{T} z_i \mathbf{x}_i / \sum_{i=1}^{T} z_i]$ ). Then, it applies a logistic regression layer (FC layer + sigmoid) to output video-level classification prediction probability. Specifically, the prediction probability for class $c \in \{1, 2, \cdots C\}$ is parameterized as $\sigma_j^c = \sigma(\mathbf{w}_c^\top \bar{\mathbf{x}}_j)$, where $\bar{\mathbf{x}}_j$ is the aggregated feature for video $j \in \{1, ..., N\}$. Suppose each video $\bar{\mathbf{x}}_j$ is i.i.d and each action class is independent from the other, the negative log-likelihood function (cross-entropy loss) is given as follows:

$$\mathcal{L}(\mathbf{W}) = -\sum_{j=1}^{N} \sum_{c=1}^{C} \left( y_j^c \log \sigma_j^c + (1 - y_j^c) \log(1 - \sigma_j^c) \right), \tag{10}$$

where $y_j^c \in \{0, 1\}$ is the ground-truth video-level label for class $c$ happening in video $j$ and $\mathbf{W} = [\mathbf{w}_1, ..., \mathbf{w}_C]$.

**Temporal Action Localization**: Let $s^c = \mathbf{w}_c^\top \bar{\mathbf{x}}$ be the video-level action prediction score, and $\sigma(s^c) = \sigma(\mathbf{w}_c^\top \bar{\mathbf{x}})$ be the video-level action prediction probability. In STPN, as $\bar{\mathbf{x}} = \sum_{t=1}^{T} \lambda_t \mathbf{x}_t$, the $s^c$ can be rewritten as:

$$s^c = \mathbf{w}_c^\top \bar{\mathbf{x}} = \sum_{t=1}^{T} \lambda_t \mathbf{w}_c^\top \mathbf{x}_t, \tag{11}$$

In STPN, the prediction score of snippet $t$ for action class c in a video is defined as:

$$s_t^c = \lambda_t \sigma(\mathbf{w}_c^\top \mathbf{x}_t), \tag{12}$$

where $\sigma(\cdot)$ denotes the sigmoid function. In MAAN, as $\bar{\mathbf{x}} = \mathbb{E}[\sum_{i=1}^{T} z_i \mathbf{x}_i / \sum_{i=1}^{T} z_i]$, according to Proposition 1, the $s^c$ can be rewritten as:

$$s^c = \mathbf{w}_c^\top \bar{\mathbf{x}} = \mathbf{w}_c^\top \mathbb{E}[\sum_{i=1}^{T} z_i \mathbf{x}_i / \sum_{i=1}^{T} z_i] = \sum_{t=1}^{T} c_t p_t \mathbf{w}_c^\top \mathbf{x}_t. \tag{13}$$

The latent discriminative probability $p_t$ corresponds to the class-agnostic attention weight for snippet $t$. According to Proposition 1 and Proposition 2, $c_t$ does not relate to snippet $t$, but captures the context of other snippets. $\mathbf{w}_c$ corresponds to the class-specific weights for action class $c$ for all the snippets, and $\mathbf{w}_c^\top \mathbf{x}_t$ indicates the relevance of snippet $t$ to class $c$. To generate temporal proposals, we compute the prediction score of snippet $t$ belonging to action class $c$ in a video as:

$$s_t^c = p_t \sigma(\mathbf{w}_c^\top \mathbf{x}_t). \tag{14}$$

We denote the $\mathbf{s}^c = (s_1^c, s_2^c, ..., s_T^c)\top$ as the class activation sequence (CAS) for class $c$. Similar to STPN, the threshold is applied to the CAS for each class to extract the one-dimensional connected components to generate its temporal proposals. We then perform non-maximum suppression among temporal proposals of each class independently to remove highly overlapped detections.

Compared to STPN (Eq. (12)), MAAN (Eq. (14)) employs the latent discriminative probability $p_t$ instead of directly using the attention weight $\lambda_t$ (equivalent to $c_t p_t$) for prediction. Proposition 2 suggests that MAAN can suppress the dominant response $s_t^c$ compared to STPN. Thus, MAAN is more likely to achieve a better performance in weakly-supervised temporal action localization.

## 3 Experiments

This section discusses the experiments on the weakly-supervised temporal action localization problem, which is our main focus. We have also extended our algorithm on addressing the weakly-supervised image object detection problem and the relevant experiments are presented in Appendix F.

### 3.1 Experimental Settings

**Datasets.** We evaluate MAAN on two popular action localization benchmark datasets, THUMOS14 (Jiang et al., 2014) and ActivityNet1.3 (Heilbron et al., 2015). **THUMOS14** contains 20 action classes for the temporal action localization task, which consists of 200 untrimmed videos (3,027 action instances) in the validation set and 212 untrimmed videos (3,358 action instances) in the test set. Following standard practice, we train the models on the validation set without using the temporal annotations and evaluate them on the test set. **ActivityNet1.3** is a large-scale video benchmark for action detection which covers a wide range of complex human activities. It provides samples from 200 activity classes with an average of 137 untrimmed videos per class and 1.41 activity instances per video, for a total of 849 video hours. This dataset contains 10,024 training videos, 4,926 validation videos and 5,044 test videos. In the experiments, we train the models on the training videos and test on the validation videos.

**Evaluation Metrics.** We follow the standard evaluation metric by reporting mean average precision (mAP) values at several different levels of intersection over union (IoU) thresholds. We use the benchmarking code provided by ActivityNet[1] to evaluate the models.

**Implementation Details.** We use two-stream I3D networks (Carreira & Zisserman, 2017) pre-trained on the Kinetics dataset (Kay et al., 2017) to extract the snippet-level feature vectors for each video. All the videos are divided into sets of non-overlapping video snippets. Each snippet contains 16 consecutive frames or optical flow maps. We input each 16 stacked RGB frames or flow maps into the I3D RGB or flow models to extract the corresponding 1024 dimensional feature vectors. Due to the various lengths of the videos, in the training, we uniformly divide each video into $T$ non-overlapped segments, and randomly sample one snippet from each segment. Therefore, we sample $T$ snippets for each video as the input of the model for training. We set $T$ to 20 in our MAAN model. The attention module in Figure 3 consists of an FC layer of $1024 \times 256$, a LeakyReLU layer, an FC layer of $256 \times 1$, and a sigmoid non-linear activation, to generate the latent discriminative probability $p_t$. We pass the aggregated video-level representation through an FC layer of $1024 \times C$ followed by a sigmoid activation to obtain class scores. We use the ADAM optimizer (Kingma & Ba, 2014) with an initial learning rate of $5 \times 10^{-4}$ to optimize network parameters. At the test time, we first reject

---

[1]https://github.com/activitynet/ActivityNet/tree/master/Evaluation

Table 1: Comparison of the proposed MAAN with four baseline feature aggregators on the THU-MOS14 test set. All values are reported in percentage. The last column is the classification mAP.

| Methods | AP@IoU | | | | | | | | | Cls mAP |
| | 0.1 | 0.2 | 0.3 | 0.4 | 0.5 | 0.6 | 0.7 | 0.8 | 0.9 | |
| --- | --- | --- | --- | --- | --- | --- | --- | --- | --- | --- |
| STPN | 57.4 | 48.7 | 40.3 | 29.5 | 19.8 | 11.4 | 5.8 | 1.7 | 0.2 | 94.2 |
| Dropout | 53.4 | 44.9 | 35.4 | 25.0 | 16.2 | 8.7 | 4.3 | 1.3 | 0.1 | 92.4 |
| Norm | 48.0 | 39.9 | 30.5 | 20.9 | 12.3 | 5.7 | 2.4 | 0.6 | 0.1 | **95.2** |
| SoftMaxNorm | 22.2 | 17.2 | 12.8 | 9.6 | 6.3 | 4.3 | 2.8 | 1.0 | 0.1 | 94.8 |
| MAAN | **59.8** | **50.8** | **41.1** | **30.6** | **20.3** | **12.0** | **6.9** | **2.6** | **0.2** | 94.1 |

classes whose video-level probabilities are below $0.1$. We then forward all the snippets of the video to generate the CAS for the remaining classes. We generate the temporal proposals by cutting the CAS with a threshold $th$. The combination ratio of two-stream modalities is set to $0.5$ and $0.5$. Our algorithm is implemented in PyTorch [2]. We run all the experiments on a single NVIDIA Tesla M40 GPU with a 24 GB memory.

## 3.2 THUMOS14 DATASET

We first compare our MAAN model on the THUMOS14 dataset with several baseline models that use different feature aggregators in Figure 3 to gain some basic understanding of the behavior of our proposed MAA. The descriptions of the four baseline models are listed below.

(1) **STPN.** It employs the weighed sum aggregation $\bar{\mathbf{x}} = \sum_{t=1}^{T} \lambda_t \mathbf{x}_t$ to generate the video-level representation. (2) **Dropout.** It explicitly performs dropout sampling with dropout probability $p = 0.5$ in STPN to obtain the video-level representation, $\bar{\mathbf{x}} = \sum_{t=1}^{T} r_t \lambda_t \mathbf{x}_t$, $r_t \sim Bernoulli(0.5)$. (3) **Normalization.** Denoted as "Norm" in the experiments, it utilizes the weighted average aggregation $\bar{\mathbf{x}} = \sum_{t=1}^{T} \lambda_t \mathbf{x}_t / \sum_{t=1}^{T} \lambda_t$ for the video-level representation. (4) **SoftMax Normalization.** Denoted as "SoftMaxNorm" in the experiments, it applies the softmax function as the normalized weights to get the weighted average aggregated video-level feature, $\bar{\mathbf{x}} = \sum_{t=1}^{T} e^{\lambda_t} \mathbf{x}_t / \sum_{t=1}^{T} e^{\lambda_t}$.

We test all the models with the cutting threshold $th$ as 0.2 of the max value of the CAS. We compare the detection average precision (%) at IoU = [0.1 : 0.1 : 0.9] and the video-level classification mean average precision (%) (denoted as Cls mAP) on the test set in Table 1. From Table 1, we can observe that although all the methods achieve a similar video-level classification mAP, their localization performances vary a lot. It shows that achieving a good video-level classification performance cannot guarantee obtaining a good snippet-level localization performance because the former only requires the correct prediction of the existence of an action, while the latter requires the correct prediction of both its existence and its duration and location. Moreover, Table 1 demonstrates that MAAN consistently outperforms all the baseline models at different levels of IoUs in the weakly-supervised temporal localization task. Both the "Norm" and "SoftmaxNorm" are the normalized weighted average aggregation. However, the "SoftmaxNorm" performs the worst, because the softmax function over-amplifies the weight of the most salient snippet. As a result, it tends to identify very few discriminative snippets and obtains sparse and non-integral localization. The "Norm" also performs worse than our MAAN. It is the normalized weighted average over the snippet-level representation, while MAAN can be considered as the normalized weighted average (expectation) over the subset-level representation. Therefore, MAAN encourages the identification of dense and integral action segments as compared to "Norm" which encourages the identification of only several discriminative snippets. MAAN works better than "Dropout" because "Dropout" randomly drops out the snippets with different attention weights by uniform probabilities. At each iteration, the scale of the aggregated feature varies a lot, however, MAAN samples with the learnable latent discriminative probability and conducts the expectation of keeping the scale of the aggregated feature stable. Compared to STPN, MAAN also achieves superior results. MAAN implicitly factorizes the attention weight into $c_t p_t$, where $p_t$ learns the latent discriminative probability of the current snippet, and $c_t$ captures the contextual information and regularizes the network to learn a more informative aggregation. The properties of MAA disallow the predicted class activation sequences to concentrate on the most salient regions. The quantitative results show the effectiveness of the MAA feature aggregator.

---

[2] https://github.com/pytorch/pytorch

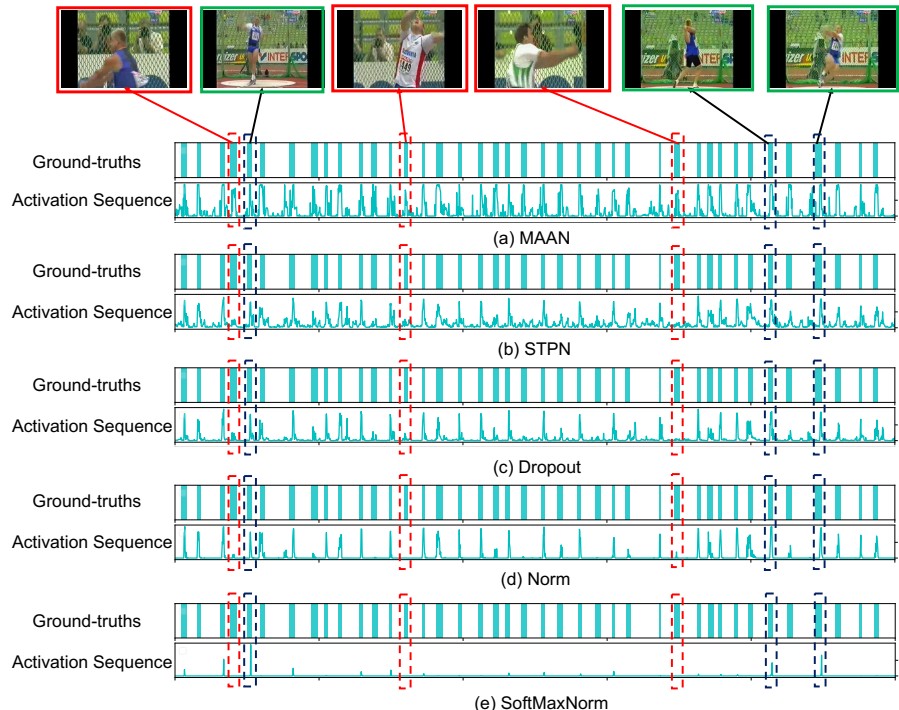

Figure 5: Visualization of the one-dimensional activation sequences on an example of the Hammer-Throw action in the test set of THUMOS14. The horizontal axis denotes the temporal dimension, which is normalized to [0, 1]. The first row of each model shows the ground-truth action segments. The second row demonstrates the predicted activation sequence for class HammerThrow.

Figure 5 visualizes the one-dimensional CASs of the proposed MAAN and all the baseline models. The temporal CAS generated by MAAN can cover large and dense regions to obtain more accurate action segments. In the example in Figure 5, MAAN can discover almost all the actions that are annotated in the ground-truth; however, the STPN have missed several action segments, and also tends to only output the more salient regions in each action segment. Other methods are much sparser compared to MAAN. The first row of Figure 5 shows several action segments in red and in green, corresponding to action segments that are relatively difficult and easy to be localized, respectively. We can see that all the easily-localized segments contain the whole person who is performing the "HammerThrow" action, while the difficultly-localized segments contain only a part of the person or the action. Our MAAN can successfully localize the easy segments as well as the difficult segments; however, all the other methods fail on the difficult ones. It shows that MAAN can identify several dense and integral action regions other than only the most discriminative region which is identified by the other methods.

We also compare our model with the state-of-the-art action localization approaches on the THU-MOS14 dataset. The numerical results are summarized in Table 2. We include both fully and weakly-supervised learning, as in (Nguyen et al., 2018). As shown in Table 2, our implemented STPN performs slightly better than the results reported in the original paper (Nguyen et al., 2018). From Table 2, our proposed MAAN outperforms the STPN and most of the existing weakly-supervised action localization approaches. Furthermore, our model still presents competitive results compared with several recent fully-supervised approaches even when trained with only video-level labels.

### 3.3 ACTIVITYNET1.3 DATASET

We train the MAAN model on the ActivityNet1.3 training set and compare our performance with the recent state-of-the-art approaches on the validation set in Table 3. The action segment in ActivityNet is usually much longer than that of THUMOS14 and occupies a larger percentage of a video. We use a set of thresholds, which are $[0.2, 0.15, 0.1, 0.05]$ of the max value of the CAS, to generate the proposals from the one-dimensional CAS. As shown in Table 3, with the set of thresholds, our implemented STPN performs slightly better than the results reported in the original paper (Nguyen

Table 2: Comparison of our algorithm to the previous approaches on THUMOS14 test set. AP (%) is reported for different IoU thresholds. Both the fully-supervised and the weakly-supervised results are listed. ("UN": using UntrimmedNet features, "I3D": using I3D features, "ours": our implementation.)

| Supervision | Methods | AP@IoU | | | | | | | | |
|---|---|---|---|---|---|---|---|---|---|---|
| | | 0.1 | 0.2 | 0.3 | 0.4 | 0.5 | 0.6 | 0.7 | 0.8 | 0.9 |
| Fully Supervised | Richard et al. (Richard & Gall, 2016) | 39.7 | 35.7 | 30.0 | 23.2 | 15.2 | - | - | - | - |
| | Shou et al. (Shou et al., 2016) | 47.7 | 43.5 | 36.3 | 28.7 | 19.0 | 10.3 | 5.3 | - | - |
| | Yeung et al. (Yeung et al., 2016) | 48.9 | 44.0 | 36.0 | 26.4 | 17.1 | - | - | - | - |
| | Yuan et al. (Yuan et al., 2016) | 51.4 | 42.6 | 33.6 | 26.1 | 18.8 | - | - | - | - |
| | Shou et al. (Shou et al., 2017) | - | - | 40.1 | 29.4 | 23.3 | 13.1 | 7.9 | - | - |
| | Yuan et al. (Yuan et al., 2017b) | 51.0 | 45.2 | 36.5 | 27.8 | 17.8 | - | - | - | - |
| | Xu et al. (Xu et al., 2017) | 54.5 | 51.5 | 44.8 | 35.6 | 28.9 | - | - | - | - |
| | Zhao et al. (Zhao et al., 2017) | 66.0 | 59.4 | 51.9 | 41.0 | 29.8 | - | - | - | - |
| Weakly Supervised | Wang et al. (Wang et al., 2017) | 44.4 | 37.7 | 28.2 | 21.1 | 13.7 | - | - | - | - |
| | Singh & Lee (Singh & Lee, 2017) | 36.4 | 27.8 | 19.5 | 12.7 | 6.8 | - | - | - | - |
| | STPN (Nguyen et al., 2018) (UN) | 45.3 | 38.8 | 31.1 | 23.5 | 16.2 | 9.8 | 5.1 | 2.0 | **0.3** |
| | STPN (Nguyen et al., 2018) (I3D) | 52.0 | 44.7 | 35.5 | 25.8 | 16.9 | 9.9 | 4.3 | 1.2 | 0.1 |
| | STPN (Nguyen et al., 2018) (ours) | 57.4 | 48.7 | 40.3 | 29.5 | 19.8 | 11.4 | 5.8 | 1.7 | 0.2 |
| | AutoLoc (Shou et al., 2018) | - | - | 35.8 | 29.0 | **21.2** | **13.4** | 5.8 | - | - |
| | **MAAN (ours)** | **59.8** | **50.8** | **41.1** | **30.6** | 20.3 | 12.0 | **6.9** | **2.6** | 0.2 |

Table 3: Comparison of our algorithm to the state-of-the-art approaches on ActivityNet1.3 validation set. AP (%) is reported for different IoU threshold $\alpha$. ("ours" means our implementation.)

| Supervision | Methods | AP @ IoU | | |
|---|---|---|---|---|
| | | 0.5 | 0.75 | 0.95 |
| Fully-supervised | Singh & Cuzzolin (Singh & Cuzzolin, 2016) | 34.5 | - | - |
| | Wang & Tao (Wang & Tao, 2016) | 45.1 | 4.1 | 0.0 |
| | Shou et al. (Shou et al., 2017) | 45.3 | 26.0 | 0.2 |
| | Xiong et al. (Xiong et al., 2017) | 39.1 | 23.5 | 5.5 |
| Weakly-supervised | STPN (Nguyen et al., 2018) | 29.3 | 16.9 | 2.6 |
| | STPN (Nguyen et al., 2018) (ours) | 29.8 | 17.7 | 4.1 |
| | **MAAN (ours)** | **33.7** | **21.9** | **5.5** |

et al., 2018). With the same threshold and experimental setting, our proposed MAAN model outperforms the STPN approach on the large-scale ActivityNet1.3. Similar to THUMOS14, our model also achieves good results that are close to some of the fully-supervised approaches.

## 4 CONCLUSION

We have proposed the marginalized average attentional network (MAAN) for weakly-supervised temporal action localization. MAAN employs a novel marginalized average aggregation (MAA) operation to encourage the network to identify the dense and integral action segments and is trained in an end-to-end fashion. Theoretically, we have proved that MAA reduces the gap between the most discriminant regions in the video to the others, and thus MAAN generates better class activation sequences to infer the action locations. We have also proposed a fast algorithm to reduce the computation complexity of MAA. Our proposed MAAN achieves superior performance on both the THUMOS14 and the ActivityNet1.3 datasets on weakly-supervised temporal action localization tasks compared to current state-of-the-art methods.

## 5 ACKNOWLEDGEMENT

We thank our anonymous reviewers for their helpful feedback and suggestions. Prof. Ivor W. Tsang was supported by ARC FT130100746, ARC LP150100671, and DP180100106.

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

# A  PROOF OF PROPOSITION 1

## A.1  PROOF OF EQUATION (3)

*Proof.*

$$\mathbb{E}\left[\frac{\sum_{i=1}^{T} z_i \mathbf{x}_i}{\sum_{i=1}^{T} z_i}\right] = \sum_{i=1}^{T} \mathbb{E}[z_i / \sum_{i=1}^{T} z_i]\mathbf{x}_i. \tag{15}$$

In addition,

$$\mathbb{E}[z_i / \sum_{i=1}^{T} z_i] = p_i \times \mathbb{E}\left[1/(1 + \sum_{k=1, k\neq i}^{T} z_k)\right] + (1 - p_i) \times 0 = p_i c_i. \tag{16}$$

Thus, we achieve

$$\mathbb{E}\left[\frac{\sum_{i=1}^{T} z_i \mathbf{x}_i}{\sum_{i=1}^{T} z_i}\right] = \sum_{i=1}^{T} c_i p_i \mathbf{x}_i = \sum_{i=1}^{T} \lambda_i \mathbf{x}_i. \tag{17}$$

□

## A.2  PROOF OF $p_i \geq p_j \Leftrightarrow c_i \geq c_j \Leftrightarrow \lambda_i \geq \lambda_j$

*Proof.* Denote $S_T = \sum_{k=1, k\neq i, k\neq j}^{T} z_k$, then we have

$$c_i - c_j = \mathbb{E}\left[1/(1 + \sum_{k\neq i} z_k)\right] - \mathbb{E}\left[1/(1 + \sum_{k\neq j} z_k)\right] \tag{18}$$

$$= p_j \mathbb{E}\left[1/(2 + S_T)\right] + (1 - p_j)\mathbb{E}\left[1/(1 + S_T)\right] - p_i \mathbb{E}\left[1/(2 + S_T)\right] - (1 - p_i)\mathbb{E}\left[1/(1 + S_T)\right]$$

$$= (p_i - p_j)\left(\mathbb{E}\left[1/(1 + S_T)\right] - \mathbb{E}\left[1/(2 + S_T)\right]\right). \tag{19}$$

Since $\mathbb{E}\left[1/(1 + S_T)\right] - \mathbb{E}\left[1/(2 + S_T)\right] > 0$, we achieve that $p_i \geq p_j \Leftrightarrow c_i \geq c_j$. Since $\lambda_i = c_i p_i$ and $\lambda_j = c_j p_j$, and $c_i, c_j, p_i, p_j \geq 0$, it follows that $p_i \geq p_j \Leftrightarrow \lambda_i \geq \lambda_j$.

□

# B  PROOF OF PROPOSITION 2

*Proof.* $\sum_{i=1}^{T} c_i p_i = \sum_{i=1}^{T} \mathbb{E}[z_i / \sum_{i=1}^{T} z_i] = \mathbb{E}\left[(\sum_{i=1}^{T} z_i)/(\sum_{i=1}^{T} z_i)\right] = 1$

When $p_1 = p_2 = \cdots = p_T$, we have $\lambda_1 = \lambda_2 = \cdots = \lambda_T$. Then inequality (4) trivially holds true. Without loss of generality, assume $p_1 \geq p_2 \geq \cdots \geq p_T$ and there exists a strict inequality. Then $\exists k \in \{1, ..., T - 1\}$ such that $c_i \geq 1/(\sum_{t=1}^{T} p_t)$ for $1 \leq i \leq k$ and $c_j \leq 1/(\sum_{t=1}^{T} p_t)$ for $k < j \leq T$. Otherwise, we obtain $c_i \geq 1/(\sum_{t=1}^{T} p_t)$ or $c_i \leq 1/(\sum_{t=1}^{T} p_t)$ for $1 \leq i \leq T$ and there exists a strict inequality. It follows that $\sum_{i=1}^{T} c_i p_i > 1$ or $\sum_{i=1}^{T} c_i p_i < 1$, which contradicts $\sum_{i=1}^{T} c_i p_i = 1$. Thus, we obtain the set $\mathcal{I} \neq \emptyset$.

Without loss of generality, for $1 \leq i \leq k$ and $i \leq j \leq T$, we have $c_i \geq 1/(\sum_{t=1}^{T} p_t)$ and $p_i \geq p_j$, then we obtain that $c_i \geq c_j$. It follows that

$$p_i/(\sum_{t=1}^{T} p_t) - p_j/(\sum_{t=1}^{T} p_t) - \left(\lambda_i/(\sum_{t=1}^{T} \lambda_t) - \lambda_j/(\sum_{t=1}^{T} \lambda_t)\right) \tag{20}$$

$$= p_i/(\sum_{t=1}^{T} p_t) - p_j/(\sum_{t=1}^{T} p_t) - (c_i p_i - c_j p_j) \tag{21}$$

$$= \left(1/(\sum_{t=1}^{T} p_t) - c_i\right)p_i - \left(1/(\sum_{t=1}^{T} p_t) - c_j\right)p_j \tag{22}$$

$$\leq \left(1/(\sum_{t=1}^{T} p_t) - c_i\right)p_i - \left(1/(\sum_{t=1}^{T} p_t) - c_i\right)p_j \tag{23}$$

$$= \left(1/(\sum_{t=1}^{T} p_t) - c_i\right)(p_i - p_j) \leq 0. \tag{24}$$

□

## C  PROOF OF PROPOSITION 3

### C.1  COMPUTATION OF $\mathbf{h}_t$

$$\mathbf{h}_t = E[\frac{\mathbf{Y_t}}{Z_t}] = \sum_{z_1, z_2, \ldots, z_t} P(z_1, z_2, \cdots z_t) \frac{\sum_{j=1}^t z_j \mathbf{x}_j}{\sum_{j=1}^t z_j} \tag{25}$$

$$= \sum_{i=0}^t \left( \sum_{z_1, z_2, \cdots z_t} \mathbf{1}\left( \sum_{j=1}^t z_j = i \right) P(z_1, z_2, \ldots, z_t) \frac{\sum_{j=1}^t z_j \mathbf{x}_j}{\sum_{j=1}^t z_j} \right) \tag{26}$$

$$= \sum_{i=0}^t \sum_{z_1, z_2, \ldots, z_t} \mathbf{1}\left( \sum_{j=1}^t z_j = i \right) P(z_1, z_2, \cdots z_t) \frac{\sum_{j=1}^t z_j \mathbf{x}_j}{i} \tag{27}$$

$$= \sum_{i=0}^t \mathbf{m}_i^t, \tag{28}$$

where $\mathbf{1}(\cdot)$ denotes the indicator function.

We achieve Eq. (26) by partitioning the summation into $t+1$ groups . Terms belonging to group $i$ have $\sum_{j=1}^t z_j = i$.

Let $\mathbf{m}_i^t = \sum_{z_1, z_2, \cdots z_t} \mathbf{1}\left( \sum_{j=1}^t z_j = i \right) P(z_1, z_2, \cdots z_t) \frac{\sum_{j=1}^t z_j \mathbf{x}_j}{i}$, and we achieve Eq. (28).

### C.2  PROOF OF RECURRENT FORMULA OF $m_i^{t+1}$

We now give the proof of the recurrent formula of Eq. (29)

$$\mathbf{m}_i^{t+1} = p_{t+1} \left( b_{i-1} \mathbf{m}_{i-1}^t + (1 - b_{i-1}) q_{i-1}^t \mathbf{x}_{t+1} \right) + (1 - p_{t+1}) \mathbf{m}_i^t. \tag{29}$$

*Proof.*

$$\mathbf{m}_i^{t+1} = \sum_{z_1, z_2, \cdots z_t, z_{t+1}} \mathbf{1}\left( \sum_{j=1}^{t+1} z_j = i \right) P(z_1, z_2, \cdots z_{t+1}) \frac{\sum_{j=1}^{t+1} z_j \mathbf{x}_j}{i} \tag{30}$$

$$= \sum_{z_1, z_2, \cdots z_t, z_{t+1}} \mathbf{1}\left( \sum_{j=1}^t z_j + z_{t+1} = i \right) P(z_1, z_2, \cdots z_t) P(z_{t+1}) \frac{\sum_{j=1}^t z_j \mathbf{x}_j + z_{t+1} \mathbf{x}_{t+1}}{i} \tag{31}$$

$$= \begin{aligned} &\sum_{z_1, z_2, \cdots z_t} \left[ \mathbf{1}\left( \sum_{j=1}^t z_j + 1 = i \right) P(z_1, z_2, \cdots z_t) p_{t+1} \frac{\sum_{j=1}^t z_j \mathbf{x}_j + \mathbf{x}_{t+1}}{i} \right] \\ &+ \sum_{z_1, z_2, \cdots z_t} \mathbf{1}\left( \sum_{j=1}^t z_j = i \right) P(z_1, z_2, \cdots z_t) (1 - p_{t+1}) \frac{\sum_{j=1}^t z_j \mathbf{x}_j}{i} \end{aligned} \tag{32}$$

$$= \begin{aligned} &\sum_{z_1, z_2, \cdots z_t} \mathbf{1}\left( \sum_{j=1}^t z_j + 1 = i \right) P(z_1, z_2, \cdots z_t) p_{t+1} \frac{\sum_{j=1}^t z_j \mathbf{x}_j + \mathbf{x}_{t+1}}{i} \\ &+ (1 - p_{t+1}) \sum_{z_1, z_2, \cdots z_t} \mathbf{1}\left( \sum_{j=1}^t z_j = i \right) P(z_1, z_2, \cdots z_t) \frac{\sum_{j=1}^t z_j \mathbf{x}_j}{i} \end{aligned} \tag{33}$$

$$= \begin{aligned} &p_{t+1} \sum_{z_1, z_2, \cdots z_t} \mathbf{1}\left( \sum_{j=1}^t z_j = i - 1 \right) P(z_1, z_2, \cdots z_t) \frac{i-1}{i} \frac{\sum_{j=1}^t z_j \mathbf{x}_j + \mathbf{x}_{t+1}}{i-1} \\ &+ (1 - p_{t+1}) m_i^t \end{aligned} \tag{34}$$

$$= \begin{aligned} &p_{t+1} \sum_{z_1, z_2, \cdots z_t} \mathbf{1}\left( \sum_{j=1}^t z_j = i - 1 \right) P(z_1, z_2, \cdots z_t) \left[ \frac{i-1}{i} \frac{\sum_{j=1}^t z_j \mathbf{x}_j}{i-1} + \frac{\mathbf{x}_{t+1}}{i} \right] \\ &+ (1 - p_{t+1}) \mathbf{m}_i^t \end{aligned} \tag{35}$$

$$= \begin{aligned} &p_{t+1} \sum_{z_1, z_2, \cdots z_t} \mathbf{1}\left( \sum_{j=1}^t z_j = i - 1 \right) P(z_1, z_2, \cdots z_t) \left[ b_{i-1} \frac{\sum_{j=1}^t z_j \mathbf{x}_j}{i-1} + (1 - b_{i-1}) \mathbf{x}_{t+1} \right] \\ &+ (1 - p_{t+1}) \mathbf{m}_i^t \end{aligned} \tag{36}$$

Then, we have

$$
\mathbf{m}_i^{t+1} = \begin{aligned} &p_{t+1}b_{i-1} \sum_{z_1,z_2,\cdots z_t} \mathbf{1}\left(\sum_{j=1}^t z_j = i-1\right) P\left(z_1, z_2, \cdots z_t\right) \frac{\sum_{j=1}^t z_j \mathbf{x}_j}{i-1} \\ &+ p_{t+1}(1-b_{i-1}) \sum_{z_1,z_2,\cdots z_t} \mathbf{1}\left(\sum_{j=1}^t z_j = i-1\right) P\left(z_1, z_2, \cdots z_t\right)\mathbf{x}_{t+1} + (1-p_{t+1})\mathbf{m}_i^t. \end{aligned}
\tag{37}
$$

Since $q_{i-1}^t = P\left(\sum_{j=1}^t z_j = i-1\right) = \sum_{z_1,z_2,\cdots z_t} \mathbf{1}\left(\sum_{j=1}^t z_j = i-1\right) P\left(z_1, z_2, \cdots z_t\right)$ we can achieve

$$
\mathbf{m}_i^{t+1} = p_{t+1}\left[b_{i-1}\mathbf{m}_{i-1}^t + (1-b_{i-1})q_{i-1}^t\mathbf{x}_{t+1}\right] + (1-p_{t+1})\mathbf{m}_i^t.
\tag{38}
$$

$\square$

## C.3 PROOF OF RECURRENT FORMULA OF $q_i^{t+1}$

We present the proof of Eq. (39)

$$
q_i^{t+1} = p_{t+1}q_{i-1}^t + (1-p_{t+1})q_i^t
\tag{39}
$$

*Proof.*

$$
q_i^{t+1} = \sum_{z_1,z_2,\cdots z_t, z_{t+1}} \mathbf{1}\left(\sum_{j=1}^{t+1} z_j = i\right) P\left(z_1, z_2, \cdots z_{t+1}\right)
\tag{40}
$$

$$
= \sum_{z_1,z_2,\cdots z_t, z_{t+1}} \mathbf{1}\left(\sum_{j=1}^t z_j + z_{t+1} = i\right) P\left(z_1, z_2, \cdots z_t\right) P(z_{t+1})
\tag{41}
$$

$$
= \sum_{z_1,z_2,\cdots z_t} \mathbf{1}\left(\sum_{j=1}^t z_j + 1 = i\right) P\left(z_1, z_2, \cdots z_t\right) p_{t+1}
\tag{42}
$$

$$
+ \sum_{z_1,z_2,\cdots z_t} \mathbf{1}\left(\sum_{j=1}^t z_j = i\right) P\left(z_1, z_2, \cdots z_t\right) (1-p_{t+1})
\tag{43}
$$

$$
= p_{t+1} \sum_{z_1,z_2,\cdots z_t} \mathbf{1}\left(\sum_{j=1}^t z_j = i-1\right) P\left(z_1, z_2, \cdots z_t\right) + (1-p_{t+1})q_i^t
\tag{44}
$$

$$
= p_{t+1}q_{i-1}^t + (1-p_{t+1})q_i^t
\tag{45}
$$

$\square$

## D RELATED WORK

**Video Action Analysis.** Researchers have developed quite a few deep network models for video action analysis. Two-stream networks (Simonyan & Zisserman, 2014) and 3D convolutional neural networks (C3D) (Tran et al., 2015) are popular solutions to learn video representations and these techniques, including their variations, are extensively used for video action analysis. Recently, a combination of two-stream networks and 3D convolutions, referred to as I3D (Carreira & Zisserman, 2017), was proposed as a generic video representation learning method, and served as an effective backbone network in various video analysis tasks such as recognition (Wang et al., 2016), localization (Shou et al., 2016), and weakly-supervised learning (Wang et al., 2017).

**Weakly-Supervised Temporal Action Localization.** There are only a few approaches based on weakly-supervised learning that rely solely on video-level class labels to localize actions in the temporal domain. Wang et al. (Wang et al., 2017) proposed a UntrimmedNet framework, where two softmax functions are applied across class labels and proposals to perform action classification and detect important temporal segments, respectively. However, using the softmax function across proposals may not be effective for identifying multiple instances. Singh et al. (Singh & Lee, 2017)

designed a Hide-and-Seek model to randomly hide some regions in a video during training and force the network to seek other relevant regions. However, the randomly hiding operation, as a data augmentation, cannot guarantee whether it is the action region or the background region that is hidden during training, especially when the dropout probabilities for all the regions are the same. Nguyen et al. (Nguyen et al., 2018) proposed a sparse temporal pooling network (STPN) to identify a sparse set of key segments associated with the actions through attention-based temporal pooling of video segments. However, the sparse constraint may force the network to focus on very few segments and lead to incomplete detection. In order to prevent the model from focusing only on the most salient regions, we are inspired to propose the MAAN model to explicitly take the expectation with respect to the average aggregated features of all the sampled subsets from the video.

**Feature Aggregators.** Learning discriminative localization representations with only video-level class labels requires the feature aggregation operation to turn multiple snippet-level representations into a video-level representation for classification. The feature aggregation mechanism is widely adopted in the deep learning literature and a variety of scenarios, for example, neural machine translation (Bahdanau et al., 2015), visual question answering (Hermann et al., 2015), and so on. However, most of these cases belong to fully-supervised learning where the goal is to learn a model that attends the most relevant features given the supervision information corresponding to the task directly. Many variant feature aggregators have been proposed, ranging from non-parametric max pooling and average pooling, to parametric hard attention (Gkioxari et al., 2015), soft attention (Vaswani et al., 2017; Sharma et al., 2015), second-order pooling (Girdhar & Ramanan, 2017; Kong & Fowlkes, 2017), structured attention (Kim et al., 2017; Mensch & Blondel, 2018), graph aggregators (Zhang et al., 2018a; Hamilton et al., 2017), and so on. Different from the fully-supervised setting where the feature aggregator is designed for the corresponding tasks, we develop a feature aggregator that is trained only with class labels, and then to be used to predict the dense action locations for test data. Different from the heuristic approaches (Wei et al., 2017; Zhang et al., 2018b) which can be considered as a kind of hard-code attention by erasing some regions with a hand-crafted threshold, we introduce the end-to-end differentiable marginalized average aggregation which incorporates learnable latent discriminative probabilities into the learning process.

## E  MARGINALIZED AVERAGE AGGREGATION

---
**Algorithm 1** Marginalized Average Aggregation
---

**Input:** Feature Representations $\{\mathbf{x}_1, \mathbf{x}_2, \cdots \mathbf{x}_T\}$, Sampling Probability $\{p_1, p_2, \cdots p_T\}$.
**Output:** Aggregated Representation $\bar{\mathbf{x}}$
Initialize $\mathbf{m}_0^0 = \mathbf{0}$, $q_0^0 = 1$, $b_i = \frac{i}{i+1}$;
**for** $t = 1$ **to** $T$ **do**
    Set $\mathbf{m}_0^t = \mathbf{0}$, and $q_{-1}^t = 0$ and $q_{t+1}^t = 0$;
    **for** $i = 1$ **to** $t$ **do**
        $q_i^t = p_t q_{i-1}^{t-1} + (1 - p_t) q_i^{t-1}$
        $\mathbf{m}_i^t = p_t \left( b_{i-1} \mathbf{m}_{i-1}^{t-1} + (1 - b_{i-1}) q_{i-1}^{t-1} \mathbf{x}_t \right) + (1 - p_t) \mathbf{m}_i^{t-1}$
    **end for**
**end for**
Return $\bar{\mathbf{x}} = \sum\limits_{i=0}^{T} \mathbf{m}_i^T$

---

## F  EXPERIMENTS ON WEAKLY-SUPERVISED IMAGE OBJECT LOCALIZATION

### F.1  MODELS AND IMPLEMENTATION DETAILS

We also evaluate the proposed model on the weakly-supervised object localization task. For weakly-supervised object localization, we are given a set of images in which each image is labeled only with its category label. The goal is to learn a model to predict both the category label as well as the bounding box for the objects in a new test image.

Table 4: Localization error on CUB-200-2011 test set

| Methods | top1 err@IoU0.5 | top5 err@IoU0.5 |
|---|---|---|
| GoogLeNet-GAP ((Zhou et al., 2016b)) | 59.00 | - |
| weighted-CAM 4x4 | 58.51 | 51.73 |
| weighted-CAM 7x7 | 58.11 | 50.21 |
| MAAN 4x4 | 55.90 | 47.60 |
| MAAN 7x7 | 53.94 | 44.13 |

Based on the model in (Zhou et al., 2016a) (denoted as CAM model), we replace the global average pooling feature aggregator with other kinds of feature aggregator, such as the weighted sum pooling and the proposed MAA by extending the original 1D temporal version in temporal action localization into a 2D spatial version. We denote the model with weighted sum pooling as the weighted-CAM model. For the weighted-CAM model and the proposed MAAN model, we use an attention module to generate the attention weight $\lambda$ in STPN or the latent discriminative probability $p$ in MAAN. The attention module consists of a 2D convolutional layer of kernel size $1 \times 1$, stride 1 with 256 units, a LeakyReLU layer, a 2D convolutional layer of kernel size $1 \times 1$, stride 1 with 1 unit, and a sigmoid non-linear activation.

## F.2 DATASET AND EVALUATION METRIC

We evaluate the weakly-supervised localization accuracy of the proposed model on the CUB-200-2011 dataset (Wah et al., 2011). The CUB-200-2011 dataset has 11,788 images of 200 categories with 5,994 images for training and 5,794 for testing. We leverage the localization metric suggested by (Russakovsky et al., 2015) for comparison. This metric computes the percentage of images that is misclassified or with bounding boxes with less than $50\%$ IoU with the groundtruth as the localization error.

## F.3 COMPARISONS

We compare our MAA aggregator (MAAN) with the weighted sum pooling (weighted-CAM) and global average pooling (CAM (Zhou et al., 2016b)). For MAAN and weighted-CAM, we pool the convolutional feature for aggregation into two different sizes, $4 \times 4$ and $7 \times 7$. We fix all other factors (e.g. network structure, hyper-parameters, optimizer), except for the feature aggregators to evaluate the models.

### F.3.1 QUALITATIVE RESULTS

The localization errors for different methods are presented in Table 4, where the GoogLeNet-GAP is the CAM model. Our method outperforms GoogLeNet-GAP by $5.06\%$ in a Top-1 error. Meanwhile, MAAN achieves consistently lower localization error than weighted-CAM on the two learning schemes. It demonstrates that the proposed MAAN can improve the localization performance in the weakly-supervised setting. Moreover, both MAAN and weighted-CAM obtain smaller localization error when employing the $7 \times 7$ learning scheme than the $4 \times 4$ learning scheme.

### F.3.2 VISUALIZATION

Figure 6 visualizes the heat maps and localization bounding boxes obtained by all the compared methods. The object localization heat maps generated by the proposed MAAN can cover larger object regions and obtain more accurate bounding boxes.

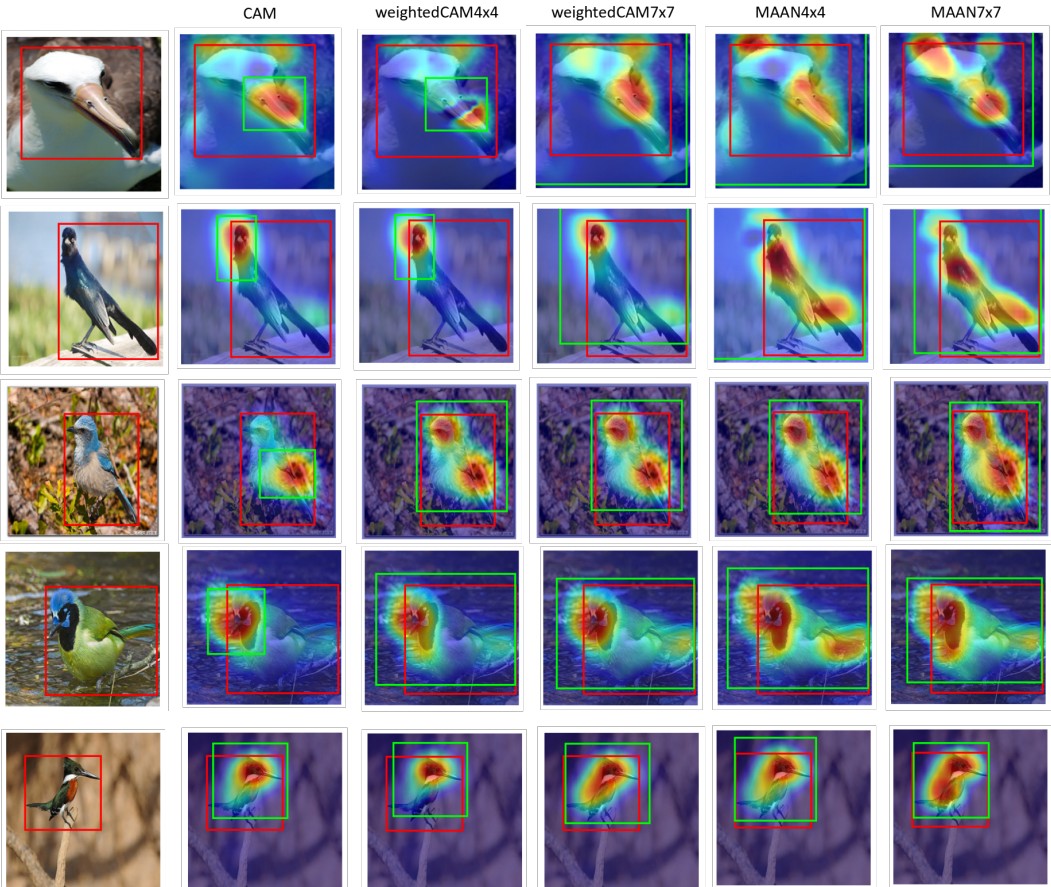

Figure 6: Comparison with the baseline methods. The proposed MAAN can locate larger object regions to improve localization performance (ground-truth bounding boxes are in red and the predicted ones are in green).

