# OpenReview forum: "MARGINALIZED AVERAGE ATTENTIONAL NETWORK FOR WEAKLY-SUPERVISED LEARNING"
_ICLR.cc/2019/Conference_

### Official Review · AnonReviewer1 · 2018-11-02
**Overly-complicated explanation of method, qualitative and quantitative results do not clearly reflect the proposed contribution.**

**Rating:** 3
**Confidence:** 3

**Review:**

In this paper the authors focus on the problem of weakly-supervised action localization. The authors state that a problem with weakly-supervised attention based methods is that they tend to focus on only the most salient regions and propose a solution to this which reduces the difference between the responses for the most salient regions and other regions. They do this by employing marginalized average aggregation to averaging a sample a subset of features in relation to their latent discriminative probability then calculating the expectation over all possible subsets to produce a final aggregation.

The problem is interesting, especially noting that current attention methods suffer from paying attention to the most salient regions therefore missing many action segments in action localization. The authors build upon an existing weakly-supervised action localization framework, having identified a weakness of it and propose a solution. The work also pays attention to the algorithm's speed which is practically useful. The experiments also compare to several other potential feature aggregators.

However, there are several weakness of the current version of the paper:

- In parts the paper feels overly complicated, particularly in the method (section 2). It would be good to see more intuitive explanations of the concepts introduce here. For instance, the author's state that c_i captures the contextual information from other video snippets, it would be good to see a figure with an example video and the behaviour of p_i and c_i as opposed to lamba_i. I found it difficult to map p_i, c_i to z and lambda used elsewhere.

- The experimental evidence does not show where the improvement comes from. The authors manage to acheieve a 4-5% improvement over STPN through their re-implemenation of the algorithm, however only have a ~2% improve with their marginalized average attention on THUMOS. I would like to know the cause in the increase over the original STPN results: is it a case of not being able to replicate the results of STPN or do the different parameter choices, such as use of leakly RELU, 20 snippets instead of 400 and only rejecting classes whose video-level probabilities are below 0.01 instead of 0.1, cause this big of an increase in results? There is also little evidence that the actual proposal (contextual information) is the reason for the reported improvement.

- There seems to be several gaps in the review of current literature. Firstly, the authors refer to Wei et al. 2017 and Zhang et al. 2018b as works which erase the most salient regions to be able to explore regions other than the most salient. The authors state that the problem with these methods is that they are not end-to-end trainable, however Li et al. 2018 'Tell Me Where to Look': Guided Attention Inference Network' proposes a method which erases regions which is trainable end-to-end. Secondly, the authors do not mention the recent work W-TALC which performs weakly-supervised action localization and outperforms STPN. It would be good to have a baseline against this method.

- The qualitative results in this paper are confusing and not convincing. It is true that the MAAN's activation sequence shows peaks which correspond to groundtruth and are not present in other methods. However, the MAAN activation sequence also shows several extra peaks not present in other methods and also not present in the groundtruth, therefore it looks like it is keener to predict the presence of the action causing more true positives, but also more false positives. It would be good to see some discussion of these failure cases and/or more qualitative results. The current figure could be easily compressed by only showing one instance of the ground-truth instead of one next to each method.

I like the idea of the paper however I am currently unconvinced by the results that this is the correct method to solve the problem.

---

> ### Author Response · Authors · 2018-11-26
> **Some clarifications about the qualitative and quantitative results**
>
>
> Thanks very much for the valuable comments and suggestions. We have clarified some questions listed below:
>
> Q: Where the improvements come from?
> Our re-implementation is not being able to replicate the results of the original STPN. Many factors may influence the performance, such as the optical flow, the RGB I3D feature and Flow I3D feature extracted with the different toolbox. The same model implemented by PyTorch and TensorFlow may also have different performance. Actually, we use 400 snippets for the re-implementation of STPN in the paper (the same as the original STPN). For SPTN, using 20 snippets is worse than using 400 snippets. The result is shown as follows:
>
> IoU threshold from 0.1:0.1:0.9
> T=20 for STPN: 43.8, 35.5, 26.0, 18.5, 10.5, 6.3, 3.4, 1.6, 0.2；
> T=400 for STPN: 57.4, 48.7, 40.3, 29.5, 19.8, 11.4, 5.8, 1.7, 0.2;
>
> We use 20 snippets for the proposed MAAN in the paper. The reason is that a small T can accelerate training, and the number of snippets has a less influence of the performance for MAAN compared with STPN.
>
> There is a typo here that we actually reject classes whose video-level probabilities are below 0.1 instead of 0.01, which we have updated in the paper.
>
> Actually, in order to reduce the influence of different factors and provide a fair comparison, we keep all the settings exactly the same between the proposed model and the re-implemented STPN model, as well as all the other compared baseline models. Under the same setting, we only change the feature aggregator in different models. The better qualitative and quantitative results compared with other baseline models empirically demonstrate that the proposed feature aggregator is the main reason for the improvement. In the paper, we also provide theoretical analysis and proof to understand and explain why the proposed model works.
>
>
> Q: Review of current literature.
> Although the model in “Tell Me Where to Look: Guided Attention Inference Network” is end-to-end trainable, it is essentially a two-stage architecture where the first stage is based on the CAM model. Our MAAN is a better alternative model for the CAM model by simply replacing the feature aggregator, which can also be served as the first stage of the model in “Tell Me Where to Look: Guided Attention Inference Network”.
> Thanks for the suggestion of the recent work W-TALC for weakly-supervised action localization, which we have missed before. We have checked the paper carefully and found that the main idea of W-TALC is a Co-Activity Similarity. The assumption is that a video pair sharing the same action label should have similar feature representations and a video pair not sharing the same action label should have a large feature difference. We do appreciate the idea. But our work is a totally different but complete story from another perspective, where the assumption, theoretical derivation, experimental results are complete and can support each other. We think both of works are beneficial to the community. Moreover, it is interesting to incorporate our method as a plug-in into these frameworks to boost the performance.
>
>
> Q: Results.
> The quantitative results show the improvement of our methods compared with the baselines in the experiments. It means that our method can bring more true positives than the false positives. We also show more qualitative results on image object localization task (Appendix F in the updated paper).

---

### Official Review · AnonReviewer3 · 2018-11-03
**Well executed paper on a reasonable idea**

**Rating:** 6
**Confidence:** 4

**Review:**

Summary
This paper proposed a stochastic pooling method over the temporal dimension for weakly-supervised video localization problem. The main motivation is to resolve a problem of discriminative attention that tends to focus on a few discriminative parts of an input data, which is not desirable for the purpose of dense labeling (i.e. localization). The proposed stochastic pooling method addressed this problem by aggregating all possible subsets of snippets, where each subset is constructed by sampling snppets from learnable sampling distribution. The proposed method showed that such approach learns more smooth attention both theoretically and empirically.

Clarity:
The paper is well written and easy to follow. The ideas and methods are clearly presented.

Originality and significance:
The proposed stochastic pooling is novel and demonstrated that empirically useful. Given that the proposed method can be generally applicable to other tasks, I think the significance of the work is also reasonable. One suggestion is applying the idea to semantic segmentation, which also shares a similar problem setting but easier to evaluate its impact than videos. Similar to (Zhou et al. 2016), you can plug the proposed pooling method on top of CNN feature map instead of global average pooling, which might be doable with the more affordable computational cost since the number of hidden units for pooling is much smaller than the length of videos (N < T).

One downside of the proposed method is its computational complexity (O(T^2)). This is much higher than the one for other feedforward methods (O(T)), which can be easily parallelized (O(1)). This can be a big problem when we have to handle very long sequences too (increasing the length of snippets could be one alternative, but it is not desirable for localization at the end). Considering this disadvantage, the performance gain by the proposed method may not be considered attractive enough.

Experiment:
Overall, the experiment looks convincing to me.

Minor comments:
Citation error: Wrong citation: Nguyen et al. CVPR 2017 -> CVPR 2018

---

> ### Author Response · Authors · 2018-11-26
> **Experiments on weakly-supervised object localization task**
>
>
> Thanks very much for the valuable comments and suggestions.
>
> We have applied the idea to weakly-supervised image object localization task. As suggested, similar to CAM (Zhou et al. 2016), we plug the proposed MAA pooling method on top of the CNN feature map instead of global average pooling. Besides compared with global average pooling, we have also compared with the weighted average pooling. The specific experimental settings and results are shown in Appendix F in the updated paper.
>
> As for the time complexity, we use 20 snippets in the training phase. At the test phase for localization, we forward each snippet i to the trained model and compute the p_i but not the lamda_i, as shown in Equation (14) (the proof is demonstrated in Proposition 2 in Section 2.2). Therefore, the time complexity at test phase is indeed O(T), which can also be easily parallelized O(1).
>
> We have corrected the citation as suggested.

---

> > ### Comment · AnonReviewer3 · 2018-12-10
> > **Response to Author Rebuttal**
> >
> > I appreciate the updated results on weakly-supervised object localization on images. Overall, I think the paper has reasonable contributions. The improvement in THUMOS14 dataset over STPN is not significant, but the results on ActivityNet look promising and the results on weakly-supervised object localization are convincing to believe that the proposed method can be generally useful to address the challenge of weakly-supervised localization where the model focuses on the most discriminative regions. For these reasons, I maintain my review score as weakly accept.

---

### Official Review · AnonReviewer2 · 2018-11-05
**Could this paper be used for other tasks beyond video action understanding?**

**Rating:** 5
**Confidence:** 3

**Review:**

This paper considers the problem of weakly-supervised temporal action localization. It proposes a marginalized average attention network (MAAN) to suppress the effect of overestimating salient regions.  Theoretically, this paper proves that the learned latent discriminative probabilities reduce the difference of responses between the most salient regions and the others. In addition, it develops a fast algorithm to reduce the complexity of constructing MAA to O(T^2). Experiments are conducted on THUMOST14 and ActivityNet 1.3.

I like the theoretical part of this paper but have concerns about the experiments. More specifically, my doubts are

- The I3D network models are not trained from scratch. The parameters are borrowed from (Carreira and Zisserman 2017), which in fact make the attention averaging very easy. I don’t know whether the success is because the proposed MAAN is working or because the feature representation is very powerful.

- If possible, I wish to see the success of the proposed method for other tasks, such as image caption generation, and machine translation.  If the paper can show success in any of such task, I would like to adjust my rating to above acceptance.

---

> ### Author Response · Authors · 2018-11-26
> **Clarifying  experimental setting  and show more experiments   on other tasks**
>
>
> Thanks very much for the comments and suggestion on other localization tasks.
>
> Actually, many works are based on the model pre-trained on other datasets like ImageNet and Kinetics (Carreira and Zisserman 2017). The compared STPN model in this paper has also used the I3D model pre-trained on Kinetics dataset. We compare the proposed MAAN with STPN and other baseline models on the exact same experimental settings.
>
> The proposed feature aggregator can be used in other weakly-supervised learning tasks. For example, we have applied the proposed method on weakly-supervised image object localization task. The experimental settings and results are shown in the Appendix F in the updated paper.

---

### Meta-Review · Area_Chair1 · 2018-12-11

**Confidence:** 3
**Recommendation:** Accept (Poster)

**Metareview:**

The paper proposes a new attentional pooling mechanism that potentially addresses the issues of simple attention-based weighted averaging (where discriminative parts/frames might get disportionately high attentions). A nice contribution of the paper is to propose an alternative mechanism with theoretical proofs, and it also presents a method for fast recurrent computation. The experimental results show that the proposed attention mechanism improves over prior methods (e.g., STPN) on THUMOS14 and ActivityNet1.3 datasets. In terms of weaknesses: (1) the computational cost may be quite significant. (2) the proposed method should be evaluated over several tasks beyond activity recognition, but it’s unclear how it would work.

The authors provided positive proof-of-concept results on weakly supervised object localization task, improving over CAM-based methods. However, CAM baseline is a reasonable but not the strongest method and the weakly-supervised object recognition/segmentation domains are much more competitive domains, so it's unclear if the proposed method would achieve the state-of-the-art by simply replacing the weighted-averaging-attentional-pooling with the proposed attention mechanism. In addition, the description on how to perform attentional pooling over images is not clearly described (it’s not clear how the 1D sequence-based recurrent attention method can be extended to 2-D cases). However, this would not be a reason to reject the paper.

Finally, the paper’s presentation would need improvement. I would suggest that the authors give more intuitive explanations and rationale before going into technical details. The paper starts with Figure 1 which is not really well motivated/explained, so it could be moved to a later part. Overall, there are interesting technical contributions with positive results, but there are issues to be addressed.